# Genomic analyses of *Mycobacterium tuberculosis* from human lung resections reveal a high frequency of polyclonal infections

Miguel Moreno-Molina [1], Natalia Shubladze [2], Iza Khurtsilava[2], Zaza Avaliani[2], Nino Bablishvili[2], Manuela Torres-Puente[1], Luis Villamayor[3], Andrei Gabrielian[4], Alex Rosenthal [4], Cristina Vilaplana [5,6,7], Sebastien Gagneux [8,9], Russell R. Kempker[10], Sergo Vashakidze[2] & Iñaki Comas [1,11✉]

Polyclonal infections occur when at least two unrelated strains of the same pathogen are detected in an individual. This has been linked to worse clinical outcomes in tuberculosis, as undetected strains with different antibiotic resistance profiles can lead to treatment failure. Here, we examine the amount of polyclonal infections in sputum and surgical resections from patients with tuberculosis in the country of Georgia. For this purpose, we sequence and analyse the genomes of *Mycobacterium tuberculosis* isolated from the samples, acquired through an observational clinical study (NCT02715271). Access to the lung enhanced the detection of multiple strains (40% of surgery cases) as opposed to just using a sputum sample (0–5% in the general population). We show that polyclonal infections often involve genetically distant strains and can be associated with reversion of the patient's drug susceptibility profile over time. In addition, we find different patterns of genetic diversity within lesions and across patients, including mutational signatures known to be associated with oxidative damage; this suggests that reactive oxygen species may be acting as a selective pressure in the granuloma environment. Our results support the idea that the magnitude of polyclonal infections in high-burden tuberculosis settings is underestimated when only testing sputum samples.

[1] Instituto de Biomedicina de Valencia IBV-CSIC, Valencia, Spain. [2] National Center for Tuberculosis and Lung Diseases of Georgia, Tbilisi, Georgia. [3] FISABIO Public Health, Valencia, Spain. [4] National Institute of Allergy and Infectious Diseases, National Institutes of Health, U.S. Department of Health and Human Services, Maryland, USA. [5] Fundació Institut Germans Trias i Pujol (IGTP), Barcelona, Spain. [6] Universitat Autònoma de Barcelona (UAB), Barcelona, Spain. [7] CIBER of Respiratory Diseases, Madrid, Spain. [8] Swiss Tropical and Public Health Institute, Basel, Switzerland. [9] University of Basel, Basel, Switzerland. [10] Department of Medicine, Division of Infectious Diseases, Emory University School of Medicine, Atlanta, USA. [11] CIBER in Epidemiology and Public Health, Madrid, Spain. ✉email: icomas@ibv.csic.es

How *Mycobacterium tuberculosis* evolves during the infection and treatment of a patient and transmits is key to understand phenomena like the spread of drug resistance. So far, the diversity of the pathogen has been mostly studied from sputum samples except for one study in the context of HIV/TB co-infection[1]. However, individual sputum samples may underestimate the true bacterial diversity within the lung as they are likely limited to reveal the coexistence of multiple *M. tuberculosis* strains in the same patient. Generally, infections involving two or more unrelated genotypes can be referred to as polyclonal (Fig. 1). Polyclonal infections of *M. tuberculosis* complicate the diagnosis and treatment of tuberculosis (TB), particularly when the infecting strains differ in their antibiotic susceptibility; which can lead to the total replacement of the susceptibility profile during treatment or to heteroresistance[2].

Polyclonal infections are also relevant to evaluate if and how an initial TB infection protects from a second infection. Recent experiments in macaques suggest that an initial infection is highly protective against reinfection and disease caused by the same *M. tuberculosis* strain[3]. However, whether the initial infection protects against heterologous challenges with a different strain is unknown. Until the advent of molecular epidemiology, the dominant view was that primary TB episodes were due to endogenous reactivation after years of latent infection. However, progression to active disease occurs mainly in the first 2 years after infection[4], suggesting that many of the episodes of TB following prolonged exposures are not due to endogenous reactivation but to reinfection. Molecular epidemiological studies also show that in high-burden countries the rate of reinfection is higher than previously recognized[5]. This is probably also true for superinfections with strains resistant to the treatment being used against the first infection.

The difficulty in identifying polyclonal infections is particularly pronounced in high-burden MDR-TB countries. Often, similar genotypes are responsible for a large proportion of recent TB transmission, making it more difficult to distinguish between these closely related strains when they coexist in the same patient[6]. In addition, in the context of drug resistance, patterns of within-host diversity in sputum cultures may be biased towards drug-resistant genotypes with high fitness. As a result, sputum-based cultures may not reflect the true extent of the pathogen diversity inside the patient's lung. For obvious reasons, studying *M. tuberculosis* directly from the lung is not usually possible. There are however a few studies, some of them based on post-mortem biopsies, which suggest that the diversity of *M. tuberculosis* within the host is higher than what can be detected in sputa[1,7]. In addition, bacterial diversity within the lungs of TB patients may affect clinical outcomes as TB drugs are known to differ in their capacity to penetrate into the different types of lung lesions[8] or into the variable immune microenvironments of individual granulomas[9–11]. Therefore, the role of within-host diversity in general and of polyclonal infections is pivotal to TB control as it has implications at many levels[12]. First, effective treatment can be compromised[13]. Second, public health interventions may need adjustments in settings with high rates of exogenous reinfection[13]. And third, there is a need to understand the role of *M. tuberculosis* strain variation in the context of new vaccines and the factors behind the limited success of the existing BCG vaccine[14,15].

The country of Georgia has a yearly incidence of 80 TB cases per 100,000 population. Out of those, 14 are MDR-TB cases (17.5%). Importantly, 12% among all new cases are MDR-TB indicating that transmission plays an important role in the epidemic of MDR-TB in Georgia[16,17]. Like in many countries of the former Soviet Union, adjunctive surgical resections are sometimes performed in patients not responding to treatment[18,19]. Utilizing these resected lung tissue samples, we studied the diversity of *M. tuberculosis* within different parts of cavitary lesions and compared it to the *M. tuberculosis* diversity seen in sputum samples of the same patients. We achieved this using culture-derived bulk sequencing of the bacteria and detecting minority variants down to 3%.

In this work, we show a high frequency of polyclonal infections and important differences between patients in terms of genetic diversity in granulomatous lesions. Some of these differences are driven by an increased mutational supply mediated by host-derived reactive oxygen species (ROS). In most patients infected with multiple strains, these strains differ in their drug resistance profiles. Furthermore, the high genetic distance observed between two strains infecting a given patient suggests an important role of strain genetic diversity in establishing a polyclonal infection. Our results represent a challenge for treatment and control of TB in the setting and highlight a possible limitation of new vaccines against TB.

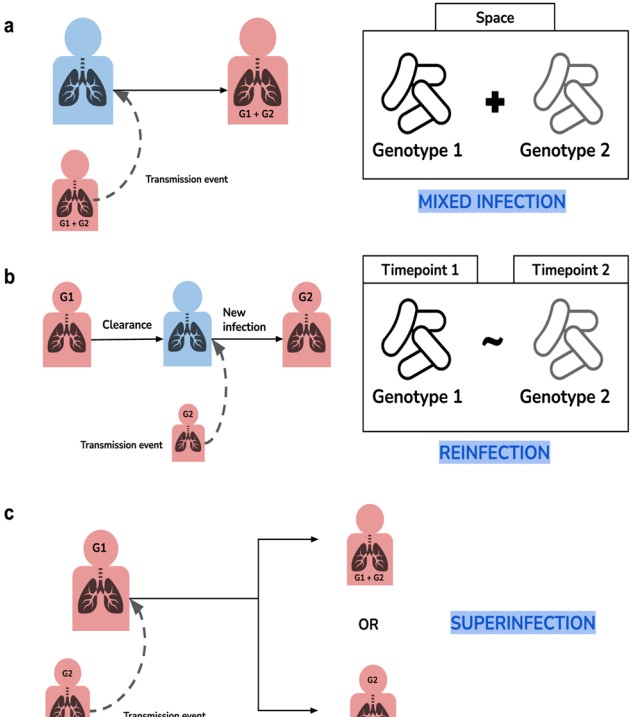

**Fig. 1 Theoretical scenarios for *Mycobacterium tuberculosis* polyclonal infections.** Blue represents healthy patients and red represents infected patients. **a** A transmission event of two strains from an infected individual to another result in two different genotypes being present in the same space or sample. **b** An infected patient on treatment clears infection and gets infected again resulting in two different genotypes present over time. **c** An already infected patient get superinfected with a different genotype. The second genotype will either coexist with the first one or replace it.

## Results

**A high frequency of polyclonal infections in MDR-TB patients from Georgia**. A total of 370 *M. tuberculosis* cultured isolates from 275 patients were included in this study (Fig. 2a). Lung surgical samples from 18 patients were also available. For nine patients, two samples were analyzed from sputum and caseum, respectively, and for nine others, we analyzed multiple samples from cavitary granulomatous lesions, samples from remote tubercular foci, and visually healthy lung tissue surrounding the cavity. In addition, for

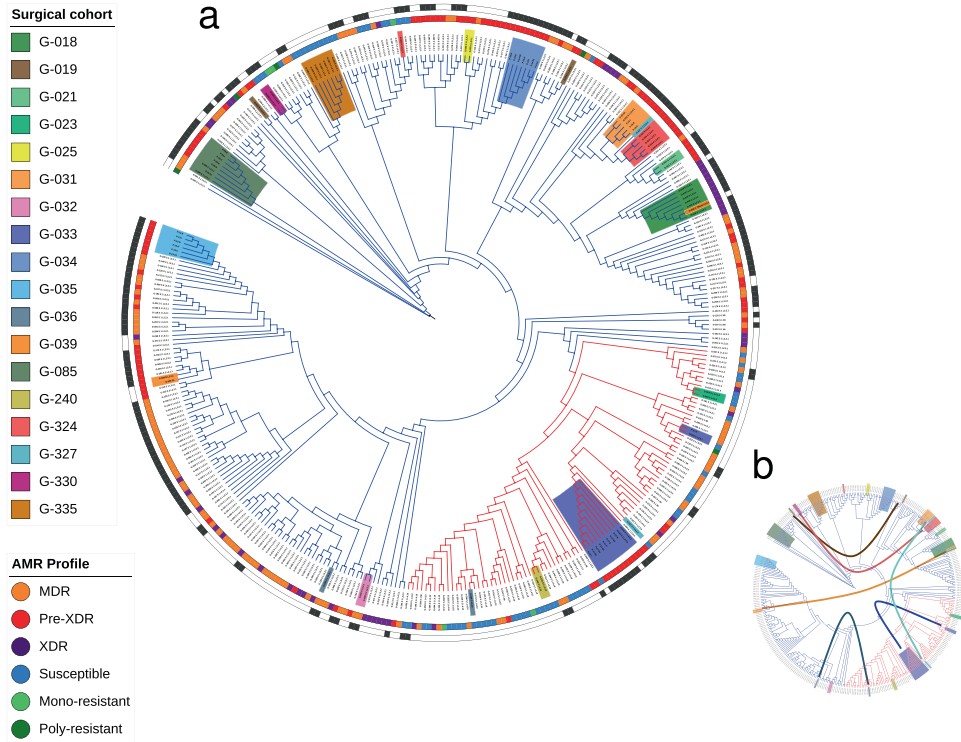

**Fig. 2 Phylogenetic diversity in Georgia and identification of polyclonal infections. a** Phylogeny of all Georgian *M. tuberculosis* isolates included in this study (L2 in blue, L4 in red). Surgery patients are highlighted in different colors. Drug resistance profiles are illustrated in the inner band (see legend) and transmission clusters represented on the outer band were estimated using a 10 SNP phylogenetic distance threshold. Branch lengths and bootstrap values not represented. **b** Curves connecting patient samples located in distant places of the phylogeny, suggesting polyclonal infections. The phylogeny and associated data can be browsed at the ITOL website (https://itol.embl.de/tree/16111121824738113158091503B).

38 patients, we analyzed two consecutive cultures from sputum samples. In terms of drug resistance profiles, patient samples were 16% pan-susceptible (44), 2% mono-resistant (3 mono-INH, 2 mono-SM, 1 mono-EMB), 2% poly-resistant (5), 41% MDR (112), 24% pre-XDR (66) and 15% XDR (42).

In regards to the sputum and lung tissue *M. tuberculosis* clinical isolates from patients undergoing adjunctive surgical resection, a genome-based phylogeny of all bulk-sequenced cultures revealed that for seven out of eighteen cases, samples from the same patient did not cluster together (Fig. 2b). This strongly suggested the presence of polyclonal infections. Patients harboring different strains in different samples become evident when considering pairwise genetic distances between these strains. It has been suggested that >10 SNPs between isolates of the same patient are unlikely to emerge by clonal diversification from a single *M. tuberculosis* genotype[20]. For all seven patients in which the isolates did not cluster together, the distance between the genotypes was much higher than 10 SNPs (range 105 to 1132 SNPs, Table 1). By contrast, in the eleven patients whose isolates clustered together, the pairwise SNP distance was between 0 and 2 SNPs (Wilcoxon Rank Sum Test; W 0, *p*-value 0.001). For example, the caseum isolate from patient G019 clustered 120 SNPs apart from the sputum isolate of the same patient. Extreme examples were patient G036 or G327, in which the sputum and the caseum isolates belonged to different lineages (L2 and L4, more than 1100 SNPs apart). For three patients, the isolates were not only located in different parts of the phylogeny but were also assigned to different transmission clusters (Supplementary Fig. 1). For patient G324, whose surgical isolates clustered with the caseum isolate from patient G327, the sputum isolates clustered with patient G005 and G312 sputum isolates. Patients G019 and G036 also had paired isolates clustering independently with other

isolates in the phylogeny instead of with each other. In total, we found 6 out of 7 polyclonal infection surgical patients involved in recent transmission clusters (Supplementary Fig. 1). These values suggest that superinfections are common among the MDR-TB population of this setting.

**TB patients undergoing lung surgery show complex infection scenarios.** For nine surgery patients, several bacterial cultures from different parts of the cavitary lesion could be analyzed in detail, including caseum (C), inner wall (I), external wall (E), remote nodule (N), and surrounding healthy tissue (H), in addition to the diagnostic sputum culture (S). Analysis of the *M. tuberculosis* genomic diversity within and around lesions showed very different patterns across patients (Supplementary Figs. 2 and 3). Detailed descriptions of the surgical cohort are provided in Supplementary Data 1.

For three surgery patients, the situation was even more complicated. For those patients, we found instances of polyclonal infections of two unrelated genotypes not only between isolates of the same individual but also within the same lesion. For instance, patient G039 harbored two genotypes in different parts of the same lesion (C, I, E, H), while the nodule (N) and sputum (S) samples harbored a single genotype. Deconvolution of the two genotypes present in the caseum isolate (see "Methods") revealed that the genotype absent from the nodule and sputum belonged to a different sublineage and was 105 SNPs apart. A similar phenomenon was observed in patient G240, where two genotypes coexisted in the granuloma center, but only one was present in the sputum isolate and clustered with another patient's sputum isolate. Finally, patient G033 also showed evidence of two genotypes belonging to different L4 sublineages, coexisting in the nodule and healthy tissue samples.

**Table 1 Summary of *M. tuberculosis* genotypes found in the surgical cohort.**

| Patient | Infection | G1 Lin. | G1 DRP | G2 Lin. | G2 DRP | Dist. | C | I | E | H | N | S |
|---|---|---|---|---|---|---|---|---|---|---|---|---|
| G018 | Clonal | L2.2.10 | XDR | – | – | 0 | G1 | G1 | G1 | G1 | G1 | G1 |
| G021 | Clonal | L2.2.10 | Pre-XDR | – | – | 0 | G1 | – | – | – | – | G1 |
| G023 | Clonal | L4.2.1 | Sus. | – | – | 1 | G1 | – | – | – | – | G1 |
| G025 | Clonal | L2.2.10 | Pre-XDR | – | – | 0 | G1 | – | – | – | – | G1 |
| G031 | Clonal | L2.2.10 | Pre-XDR | – | – | 2 | G1 | G1 | G1 | – | – | G1 |
| G032 | Clonal | L2.2.9 | MDR | – | – | 0 | G1 | – | – | – | – | G1 |
| G034 | Clonal | L2.2.10 | Pre-XDR | – | – | 0 | G1 | G1 | G1 | G1 | G1 | G1 |
| G035 | Clonal | L.2.2.9 | Pre-XDR | – | – | 0 | G1 | G1 | G1 | G1 | G1 | G1 |
| G085 | Clonal | L2.2.10 | Pre-XDR | – | – | 2 | G1 | G1 | G1 | G1 | G1 | G1 |
| G330 | Clonal | L2.2.10 | Sus. | – | – | 0 | G1 | – | – | – | – | G1 |
| G335 | Clonal | L2.2.10 | Sus. | – | – | 0 | G1 | G1 | G1 | G1 | G1 | G1 |
| G033 | Polyclonal | L4.3.3 | Pre-XDR | L4.2.1 | Sus. | 611 | G1 | G1 | G1 | M | M | G1 |
| G039 | Polyclonal | L2.2.9 | Pre-XDR | L2.2.10 | Pre-XDR | 105 | M | M | M | M | G2 | G2 |
| G240 | Polyclonal | L4.8 | Sus. | L4.8 | Sus. | 162 | M | – | – | – | – | G1 |
| G019 | Polyclonal | L2.2.10 | MDR | L2.2.10 | Pre-XDR | 120 | G1 | – | – | – | – | G2 |
| G036 | Polyclonal | L2.2.9 | MDR | L4.8 | Sus. | 1132 | G1 | – | – | – | – | G2 |
| G324 | Polyclonal | L2.2.10 | Sus. | L2.2.10 | Pre-XDR | 128 | G1 | G1 | G1 | G1 | G1 | G2 |
| G327 | Polyclonal | L4.3.3 | Sus. | L2.2.10 | Pre-XDR | 1116 | G1 | – | – | – | – | G2 |

Lineage (Lin.) and drug resistance profile (DRP) of the different genotypes (G1, G2), as well as their genetic distances (Dist.) are shown.
Genotype locations are indicated with: *G1* genotype 1, *G2* genotype 2, *M* mix, – no sample available.
*C* caseum, *I* inner wall, *E* external wall, *H* healthy tissue, *N* nodule, *S* sputum.

Taken together, our results show that seven out of the eighteen surgery patients (39%) showed evidence of infection by two phylogenetically unrelated genotypes, either in the same sample or in separate samples (Table 1). To compare the high percentage of polyclonal infections in our surgical dataset with the percentage expected when using only a single sputum diagnostic sample, we scanned for possible polyclonal infections in 218 sputum culture-positive patients from the country of Georgia. As only one sample per case was available we could only scan for cases of mixed infection. We used two different methods to identify the co-existence of two different genotypes in single sputum samples, including a phylogenetic-based approach specifically designed for this analysis (see "Methods"). In total, we identified 11/218 (5%) likely cases of co-existence in these patients based on a single sputum sample. Supporting this result, when we only took into consideration the sputum isolates from surgery patients no mixed infections were detected. This is in contrast with findings in surgical samples where we identify three cases in which two genotypes co-existed (17% in surgical samples vs. 5% in sputum samples from the general population; chi-square 4.39, *p*-value = 0.036). To increase the power to detect additional polyclonal infections in the general population, we analyzed a set of 38 patients with two consecutive sputum samples (Supplementary Table 1) collected from days to months apart (range 1–595 days). In total, we identified seven patients out of the 38 (18.4%) with evidence of polyclonal infection at some point during their TB disease. This was lower than the frequency of polyclonal infections in the surgical samples (39%) but higher than in the single diagnostic sputum samples (5%). Despite the study limitations, this suggests that the real extent of polyclonal infections cannot be accurately estimated from a single sputum sample.

We chose patient G039 to illustrate how complex infection and transmission patterns can be in a high-burden MDR-TB setting. This patient showed a polyclonal infection of two genotypes in the caseum, the inner and external wall as well as in the healthy tissue, but not in the remote nodule or sputum (Fig. 3a). A detailed analysis of the SNP frequencies suggested that there were two genotypes coexisting in the caseum at a ratio of 80:20 (genotype1:genotype2). Tracking of variable positions across samples of the patient revealed that this ratio varied. The proportion of genotype 1 decreased as we moved out of the caseum until reaching a ratio of nearly 50:50 in the external walls and healthy tissue. The situation was reversed in the remote nodule where genotype 2 was fixed at 100% frequency. The sputum of the patient also contained genotype 2 at a 100% frequency. Deconvoluting these two genotypes were challenging as both belonged to the same lineage 2 sublineages. We could extract the genetic base of genotype 2 from the nodule as it represented 100% of the culture but genotype 1 always existed as a mixture with genotype 2 in the other samples. To deconvolute genotype 1, we assigned as a fixed SNP any position that was in the caseum sample at above 75% frequency, thus reflecting the expected frequency of SNPs associated with genotype 1. Comparison with genotype 2 corroborated that both genotypes belonged to the same lineage but different sublineages (2.2.9 and 2.2.10), being only 105 SNPs apart. The frequencies of genotype 1 and genotype 2 correlated with resistance mutations to several antibiotics at the corresponding frequencies (Fig. 3b, c). Several of the resistance mutations were the same in both genotypes, including those for isoniazid, rifampicin, ofloxacin, and kanamycin. The combined frequency of different resistance mutations to ethambutol and streptomycin across the two genotypes resulted in 100% of the culture is resistant to those drugs. For other drugs like pyrazinamide, resistance mutations were only found in genotype 1.

**Patient follow-up treatment can be compromised by the polyclonal infection.** To analyze the role of polyclonal infections in drug resistance, we used the available DST results for the surgical cohort's samples. First, we explored whether there was variability within a patient in phenotypic DST results. In 13 out of the 18 surgical patients, the DST results did not change across sites even when the sputum sample was taken into account. In five patients, the DST profiles differed across samples. Sometimes the difference involved only one drug like OFX in G019 but on other occasions like patient G327, the DST profile was fully reversed when comparing the caseum and the sputum sample (from pre-XDR to pansusceptible in this case). In other instances, we had polyclonal infections like patient G033, in which the mix between the two genotypes in H and N samples (susceptible + pre-XDR) resulted in an overall profile of pre-XDR masking the

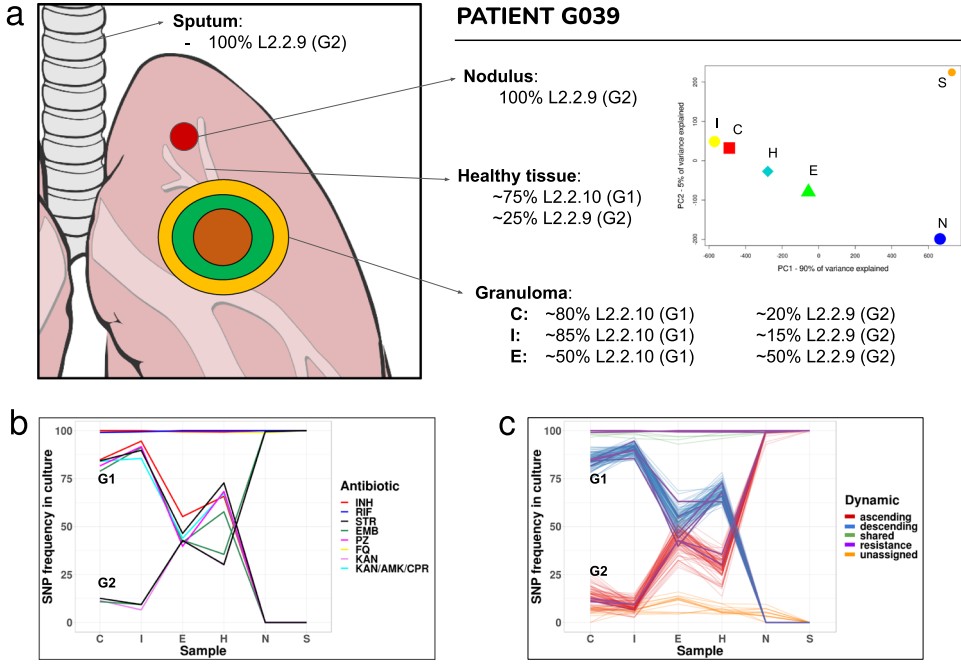

**Fig. 3 Dynamics of polyclonal infection with two different strains across the granuloma. a** Descriptive summary of patient G039 genotypes frequencies and distribution across the lesion. The PCA graph illustrates their separation: while the *X*-axis explains ~90% of the variance, leaving N and S apart from the rest, the *Y*-axis only explains ~5%, mainly defined by low-frequency variants not shared by N and S (due to reading depth differences). **b** Antibiotic resistance-associated mutations across surgical and sputum samples for the two co-existing genotypes. **c** Deconvolution of the two genotypes by frequency patterns clustering. Resistance-associated variants from (**b**) are represented in purple. Unassigned subpopulations (in orange) are low-frequency variants that we cannot assign to any of the two genotypes and shared fixed variants (in green) are common to both genotypes. C caseum, I inner wall, E external wall, H healthy tissue, N nodule, S sputum. Source data are provided as a Source Data file.

second genotype. Although the number of patients was low, three out of five patients with discrepancies in DST vs. genomic prediction had a polyclonal infection, suggesting that these infections can mislead individual treatment of MDR-TB patients. For a comparison of phenotypic and genotypic results as well as identification of novel drug resistance markers see Supplementary Note 2.

Another complication might result from the replacement over time of one strain by another, which may compromise the treatment if no additional DST is done during follow-up. In support of this notion, in all four patients harboring different strains in temporally separated samples (Table 1), the DST profile changed for several drugs. This included cases that were susceptible in the first isolate but MDR/XDR in the second isolate or vice versa, like patients G324 or G327. This data shows that treatment based on the results from one diagnostic sample can be misleading and suggests the need for sequential testing of isolates under programmatic conditions, especially in settings with a high MDR-TB burden such as the country of Georgia.

**Diversity within and around lesions reflects host microenvironment pressures**. For those patients with both surgical and sputum samples and with no polyclonal infection detected, we analyzed the *M. tuberculosis* genetic diversity across their samples. The overall diversity of *M. tuberculosis* within patients was well represented in sputum samples when compared to the caseum sample (correlations >90%). We then calculated the number of bacterial SNPs exclusive to each sample. These are SNPs that have accumulated since the divergence of the sample from the closest isolate in the phylogeny. In general, patients with a high number of exclusive SNPs in the lesions also had a high number of SNPs in the sputum ($R^2 = 0.9623$ between cavity centers and corresponding sputa using exclusive SNPs and

excluding G021 which showed a high diversity only in sputum). Thus, the general assumption that the *M. tuberculosis* diversity in sputum samples is only a subset of the within-patient diversity is not always true. Indeed, we saw patients in which the sputum harbored exclusive SNPs not found in the surgical samples and vice versa. TB patients widely differed in the amount of within-host MTB genomic diversity, with approximately half of the patients harboring almost no MTB diversity (e.g., G031, G034, G035 with no low-frequency SNPs; Fig. 4a and Supplementary Fig. 3), while others showed a large diversity across sputum and surgical samples (e.g., G023, G025 with >600 low-frequency SNPs; Fig. 4a and Supplementary Fig. 3). However, the comparison between sputum and surgery samples must be taken with caution as the sputum and surgical samples were obtained at different time points and may also reflect the sampling of different lesions.

Several selective forces, including the host response and antibiotic treatment, may affect the diversity within granulomas from the time of diagnosis to the time of surgery. It has been suggested that the host immune system may exert a mutational pressure on the infecting bacteria through the production of ROS. This has been demonstrated in single colony analyses of cultured sputum samples showing that an elevated mutational supply can be identified in immunocompetent individuals but not in HIV-positive individuals[21]. A hallmark of an elevated mutational rate due to ROS is a mutational signature associated with oxidative damage (increased changes C > T and G > A)[22]. It could be expected that such a ROS mutational signature was amplified in surgical samples compared to sputum. As our study was not based on single-colony sequencing but on bulk sequencing, we reasoned that identifying variants present in a few colonies in culture should roughly correlate with variants at very low frequency in bulk culture sequencing. Thus for this analysis, we

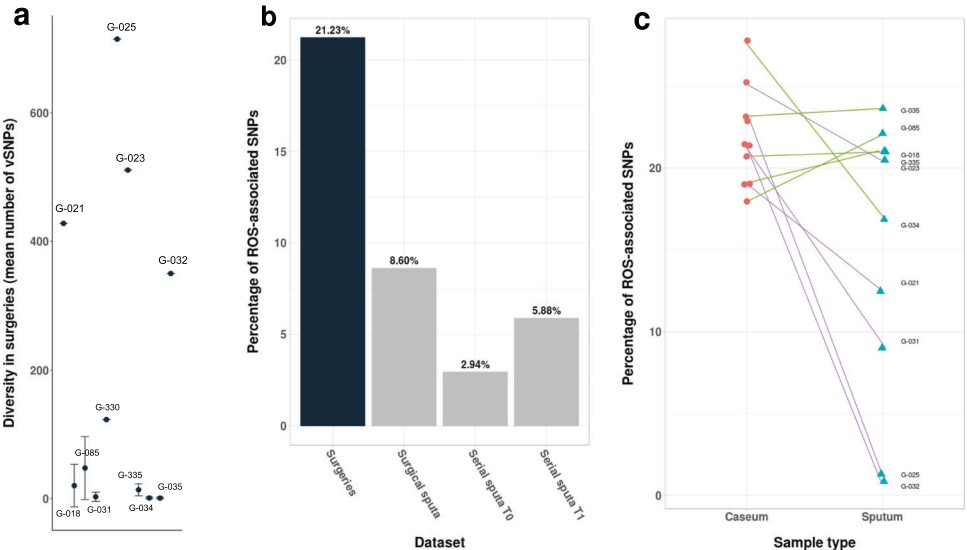

**Fig. 4 Impact of the granuloma microenvironment on within-host diversity. a** Average diversity of each patient's surgical samples (measured by a number of vSNPs and excluding polyclonal infections, $n =$ number of independent surgical samples available for each patient, range 1–3). Data are presented as mean values ± SD. **b** Pooled comparisons of ROS signature in the different datasets. Categories include surgery specimens, sputum samples from surgery patients, and serial sputum samples from the same patient. **c** Individual values of caseum and sputum from surgery patients. Purple lines connect those patients whose differences are statistically significant by the two-tailed $X^2$ test. Source data are provided as a Source Data file.

focused on variants with frequency lower than 5% for samples with enough depth of coverage and excluding patients showing polyclonal infection as frequencies in those cases are not straightforward to interpret (total eligible patients $n = 10$). Our analysis revealed that, as hypothesized, surgical samples showed a stronger ROS mutational signature compared to the sputum samples from the same patients (21% vs. 8.6%, pooled analysis, Fig. 4b; chi-square 14.5, $p$-value < 0.0001). As the sputum and surgical samples were obtained at different time points, we also analyzed patients with serial sputum samples to determine the effect of time on the diversity observed in sputum. While there was an effect of time (S1 3% vs. S2 5.9%; chi-square 22.3 $p$-value < 0.0001), the magnitude of the ROS signature in serial samples was similar to that seen in sputum samples from surgical patients (Fig. 4b) but significantly lower than in the surgical samples (chi-square $p$-value < 0.0001 for all comparisons against surgical samples). Some patients had stronger ROS signatures than others (Fig. 4c; 5 out of 10 significant at the individual level, G021, G023, G025, G031, G032; chi-square $p$-value < 0.05). This is in accordance with results by Liu and colleagues[21], where 4 out of 18 cases showed a significant increase in ROS-associated mutations. Thus, our results suggest that the granuloma microenvironment can increase the mutational supply for transitions during the within-host evolution of *M. tuberculosis*.

## Discussion

Studies on within-host diversity of *M. tuberculosis* are based on cultured samples and very few on non-sputum samples, limiting our capacity to understand diversity patterns at the site of infection. In this study, we analyzed an 18-patient cohort, most of the MDR to XDR-TB cases, with available lung surgical samples from Tbilisi (Georgia). We performed bulk sequencing of cultured bacteria from the surgical resections and sputum isolates of every patient, finding an elevated percentage of polyclonal infections with an impact on drug resistance in the surgical cohort (39%), which cannot be attained by single sputum samples. In addition, we demonstrate that patterns of genetic diversity differ within lesions and across patients suggesting a role of host immune microenvironments.

The surgical cohort showed seven out of the eighteen patients (39%) with evidence of infection by two phylogenetically unrelated genotypes, either in the same sample or in separate samples. This represents a high percentage when compared to the 5% found in single sputum patients of our dataset and percentages described by others[2]. Three surgical patients showed true mixed infections (multiple genotypes co-existing) and four are likely the results of superinfections due to multiple transmission events. The high rate of polyclonal infections in these patients suggests deficiencies in infection control in the setting[23] and agrees with epidemiological and model data showing that repeated exposure to infection as seen in high-burden settings increases the risk of reinfections, disease progression, and outbreaks[23,24]. In addition, polyclonal infections in the surgical cohort usually involve strains with different drug resistance patterns (5 out of 7 cases), a fact that can hamper successful treatment if not assessed correctly[25]. There are several studies showing that infection with multiple strains resulted in a poor treatment outcome since the presence of any undetected resistance during standard treatment can propel the acquisition of further drug resistance[13]. Importantly, in our results, polyclonal infections change, and many times fully revert, the drug resistance profile of the patients. Thus, follow-up DST samples should be implemented in these settings for better patient management.

Figure 1 shows the theoretical scenarios to explain the natural history of infection for these patients. The fact that in several cases the second isolate is in a transmission cluster suggests that superinfection is one of the most common mechanisms. However, we cannot be sure if two strains were already infecting at baseline while only one is detected in sputum as we do not have access to lung samples before treatment. Similarly, we cannot test the hypothesis suggested from mice experiments that a secondary infection drives the progression of an asymptomatic primary infection via immune response or by expression of resuscitation-promoting factors[13,26]. However, our results contribute to the evidence from mice data[26] (but not from macaques[3]) that reinfection with a second strain can be common after a primary infection under certain circumstances and calls to re-evaluate the natural history of TB in settings with the rampant transmission.

The extent of polyclonal infections also has consequences to understand the protective potential of vaccines. Recent in vivo reinfection experiments suggested that the first episode of TB was more protective against a second episode than previously thought[6]. However, our results suggest that in a high-burden setting this might not be the case, and future studies designed to corroborate this observation will be needed. Differences may be related to how well the experimental set-up recapitulates natural infection in such settings. In the experiments on macaques, the same strain was used for re-challenge[3]. By contrast, in our dataset we frequently observe different strains, usually from different (sub-)lineages, infecting the same patient. In fact, we did not see polyclonal infections with strains less than 100 SNP apart despite our phylogenetic approach being designed to identify cases up to 20 SNP apart. This raises the possibility that previous infection might indeed protect against superinfection with the same or a very similar strain, as observed in the macaque model. The differences with animal experiments may also be related to the nature of the re-challenge, as in TB transmission hotspots, the exposure to second infections is recurrent[27]. However, co-morbidities and social determinants may also play an important part in the susceptibility of the host to second infections. Lastly, many clinical trials and vaccine studies use sputum as their basis for distinguishing between relapse and reinfection and our results suggest that it may be a poor correlate to ascertain lung polyclonal infection.

Beyond multiple genotypes, analysis of the genomic diversity within and around lesions showed very different patterns across patients. Some showed almost no diversity while others were highly diverse. In some cases (notably G-018) the distribution of drug resistance variants suggests gradients of antibiotic concentration as having been described in recent literature[8,28,29]. The fact that this pattern is not observed in all patients, may be explained by different pharmacodynamics, but also by the time difference until resection between patients as the granuloma walls get hardened over time. This would limit the penetration of drugs inside, which could affect selective pressure and thus the diversity we are able to recover from genomic data[8]. But drug pressure is not the only selective force that the pathogen encounters during infection. It is known that different immune microenvironments exist within granulomas[30]. Chief among macrophage defense mechanisms is the production of reactive oxygen and nitrogen species[31]. Looking at sputum cultures, it has been proposed that an unintended effect of ROS is to increase the mutation rate of the bacteria[21]. Our data corroborate those results showing that the mutational signature of ROS, i. e., increased number of transitions, was overall enriched in surgery samples compared to sputum samples. A recent study suggests that transition bias is linked to the acquisition of drug resistance variants[32]. It is thus tempting to link ROS-increased mutational supply to an accelerated acquisition of drug resistance mutations. Combined analyses of the bacterial population in the lung during infection in humans and relevant animal models[33], drug penetration and pharmacodynamics[28] and lesion imaging[29] will help to better understand the bacterial population diversity driven by host pressures, how it is linked to the emergence and selection of drug-resistant subpopulations and, ultimately, to relapses and treatment failure in clinical settings.

Our results are necessarily limited by the characteristics of the epidemiological setting and the patients undergoing surgery. Most of the patients who underwent surgery had already been diagnosed at least as MDR-TB and did not respond to treatment. This population is characterized by substantial exposure to multiple antibiotics, a higher frequency of prior TB disease, the frequent presence of cavitary disease (associated with poor antibiotic penetration), and sometimes prolonged hospitalization that can increase the risk of superinfection. In addition, diagnostic sputum and surgical samples have several months time difference, which limited some of our analyses, but we tried to assess the effect of sampling time by comparing our surgical dataset with the serial sputa dataset (Supplementary Table 1) in which the time difference between samples is similar overall. Also, even though we had access to surgery samples, analyses were done on cultured samples and not directly on the surgery or sputum sample which can impose different biases[34]. Finally, a possible limitation of this study would be cross-contamination as an explanation for the high frequency of polyclonal infections, although we think it is highly unlikely for a number of reasons. First, sample collection dates and processing dates from all patients are not close in time thus they have not shared the same space. Second, sample homogenization is carried out in a closed special, disposable tubes so that samples cannot be mixed. Third, genotypes match their DST phenotypes in nearly all cases, arguing against a general contamination problem. Lastly, not all polyclonal infections are in a transmission cluster and thus they do not match any other strain processed in the laboratory.

In summary, we have shown that surgical lung resections from TB patients reveal a more complete picture of the within-patient diversity of *M. tuberculosis* compared to sputum samples. The high frequency of polyclonal infections found may be related to the nature of our patient population, yet this allowed us to study complex infection scenarios with the potential to confound diagnosis and/or DST results, as in many cases the two genotypes involved in the infection had different drug resistance profiles. The surgery patients are often in a transmission cluster suggesting that either there are uncontrolled hotspots of transmission shared by MDR-TB patients, there is increased host susceptibility to reinfection by different genotypes, or both. The fact that polyclonal infections usually involve strains of different lineages or sub-lineages suggests that vaccine preclinical models must take into account the genetic diversity of the bacteria to assess protection. Finally, as observed in culture-based sputum studies[1,25,35], there are profound differences in diversity between patients and here we show that those differences can also be seen across regions of the cavitary lesions and between patients. In some patients, differences are driven by a mutational signature associated with ROS and by treatment and suggest a link between immune and drug resistance selective pressures. Overall, our results exemplify a challenge for tuberculosis treatment and control, highlighting our knowledge gap on the natural history of the disease in certain settings and the need for better patient follow-up in high burden MDR-TB areas to halt the spread of resistant strains.

## Methods

**Sampling**. We included 275 TB patients in this study (257 with diagnostic sputum and 18 with paired sputum and lung tissue samples, Supplementary Data 1) with a total of 370 *M. tuberculosis* clinical isolates from between 9 January 2013 and 20 March 2018 (see Fig. 5 for details on the surgical cohort). Sputum samples were collected according to standard clinical practice. The surgical samples used in this study were acquired within the framework of observational clinical study NCT02715271, which is ongoing as of this publication date but no longer recruiting, and results are reporting on primary outcome #3. Consent to publish clinical information potentially identifying individuals was obtained. The main indication for lung surgery was the persistence of abnormal lung lesions (predominantly cavities) identified on chest X-ray (13/18), both in DS (3/13) and MDR/XDR-TB (10/13), despite good treatment adherence. Additional indications for surgery included treatment failure (2.9%), complications of TB related to pulmonary hemorrhage, spontaneous pneumothorax, or empyema[36]. Immediately after lung resection, samples were removed from the following zone of obtention: cavity center (C), cavity internal wall (I), cavity external wall (E), visually healthy tissue around the cavity (H) and nodule (N), placed in sterile tubes and sent to the National Center for Tuberculosis and Lung Disease microbiological laboratory for processing. In total, we collected 219 single sputa, 38 pairs S1–S2 (serial sputa), 9 pairs C–S (caseum–sputum), and 9 complete surgical sets (C, I, E, H, N, S).

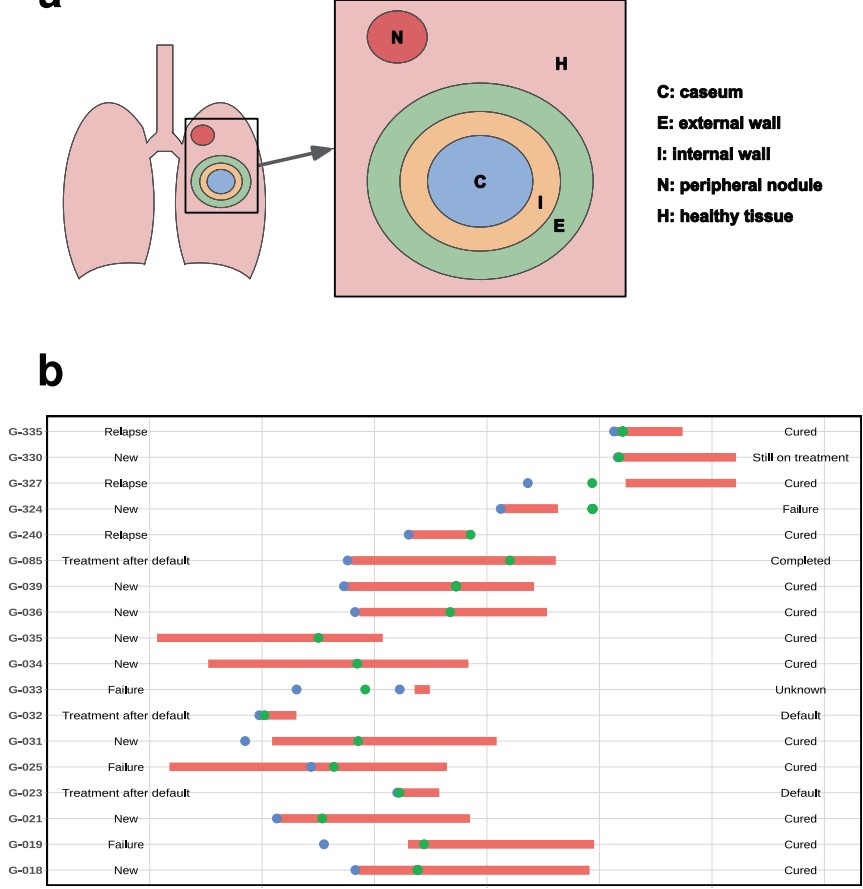

**Fig. 5 Details on the surgical cohort. a** Sampling sites for the surgical cohort. **b** Timeline of key events for the surgical cohort. Treatment periods are depicted in red, sputum samples date in blue points, and surgical samples date in green points. Starting case definition and treatment outcome are also represented. Source data are provided as a Source Data file.

**Culture and drug susceptibility testing**. Resected tissue samples were homogenized using Minilys homogenizer, placed in both LJ and BACTEC 960 MGIT™ culture, and processed as per the manufacturer's instructions. Sputum samples were also cultured in both media. All clinical isolates were confirmed for *M. tuberculosis* complex (MTBC) using the standard microbiological method[37,38]. All isolates underwent drug susceptibility testing (DST) for first and second-line drugs in both solid (LJ) with increasing drug concentrations and liquid media BBL® MGIT™, as recommended by the manufacturer.

**DNA extraction and sequencing**. For both sputum and resected tissue cultures, after reaching sufficient bacterial growth in the LJ medium, plates were thoroughly scraped to maximize diversity recovery for bulk sequencing. DNA extraction was performed by the standardized protocol in the BSL-2+ laboratory from the AFB positive culture. In short, bacteria were heated at 80 °C for 20 min, centrifuged, and resuspended in TE buffer. Lysozyme, SDS, and proteinase K were added in sequential incubations. DNA was precipitated using chloroform-isoamyl alcohol and isopropanol and resuspended in TE buffer[39]. When there was no sufficient growth in LJ, the CTAB/chloroform method was used for DNA extraction from 1 mL of MGIT culture. The quality and quantity of DNA were analyzed with the help of spectrophotometers (Thermo Scientific NanoDrop 2000 and Qubit 3.0). Extracted DNAs were sent for whole-genome sequencing to Broad Institute, C-PATH, TGen North, or Tuberculosis Genomics Unit (IBV-CSIC). The average sequencing depth for this study was 141X. The average sequencing depth for surgical samples was 267X.

**Bioinformatic analysis**. Read preprocessing was done using *fastp*[40] to scan reads and trim low-quality ends with a mean window quality <20. We then used Kraken[41] to taxonomically classify reads by means of a custom database, only keeping MTBC sequences to avoid false variants arising due to contaminant DNA. Filtered reads were mapped with BWA[42] to a predicted MTBC ancestor reference sequence[43] using default parameters, and processed using samtools[44] and Picard[45].

After that, we scanned for optical and PCR duplicates to remove them, as this helps to reduce the number of artifactual variants in low-frequency ranges. Variant calling for sputum samples was carried out using the software and parameters from the calling module of the pipeline validated for sputum sample cultures at the IBV-CSIC (available at https://gitlab.com/tbgenomicsunit/ThePipeline).

For a robust variant calling in surgery samples, we used three different variant callers (VarScan2[46], GATK's HaplotypeCaller[46,47], LoFreq[48]) and integrated SNPs reported by at least two of them to get a high-confidence list of low-frequency variants. VarScan2 was run with parameters "*pileup2snp sample.pileup—min-coverage 20—min-reads2 4—min-avg-qual 20—min-var-freq 0.01—min-freq-for-hom 0.9—strand-filter 1*", GATK was run with parameters "*-T HaplotypeCaller -R ref.fasta -I sample.bam -o sample.vcf—min-base-quality-score 20 -ploidy 1*" and LoFreq was run with parameters "*call-parallel—pp-threads 12 -f ref.fasta -o sample.vcf sample.bam*" and "*filter -i sample.vcf -v 20 -A 0.01 -Q 20 -o filtered.vcf*".

From the initial list of variants, we applied a mapping filter that discarded variants that arose in repetitive genomic regions like the PE/PPE families or phages[49]. To establish a threshold that discarded additional false variants, we performed a synthetic read simulation. Using the ART software package[50], we got the quality distribution and error profiles of our samples (using *art_profiler_illumina* with default parameters), and simulated 100 sequencing runs with the data (*art_illumina -p −1 profile.txt -2 profile.txt -na -iref.fasta -l 150 -f 1000 -m 280 -s 137 -o out.fas*). By analyzing simulations with the same pipeline, we defined a ~3% minimum frequency threshold to validate a variant in surgery samples. In addition, we extended our mapping filter to new regions that showed high-frequency SNPs in the simulations and were due to systematic mapping errors to the predicted ancestor, especially in Lineage 2 strains (see Supplementary Data 2 for a list of discarded genomic features).

**Phylogenetic and population genetics analyses**. A maximum-likelihood phylogeny of the 370 samples in the dataset was constructed using IQ-TREE 2[51] from an alignment with gaps and no resistance SNP positions (run parameters: "-m GTR

-bb 1000", meaning the use of a general time-reversible model with unequal rates and unequal base frequencies, plus 1000 ultrafast bootstrap replicates). In addition, an NJ phylogeny was constructed from the same alignment using MEGA[52] and phylogenetic Hamming distances between pairs of samples were extracted by parsing the branch lengths from the tree using Python's ete3 toolkit[53].

We parsed SNP files by means of a custom Python script to obtain summary tables for each patient and a general one collecting information about the different numbers of fixed (fSNPs) and variable SNPs (vSNPs). All figures illustrating diversity across sites by the patient were produced using R[54] and the ggplot2 package[55]. For comparisons within a patient's samples, we analyzed every SNP in their samples and calculated the differences. This was performed by means of an in-house Python script that works with the SNP files for every patient, analyzing them sequentially and establishing the population dynamic for every variant based on their presence/absence and frequency, obtaining stable, sporadic, ascending, or descending dynamics. The data were plotted using R and the ggplot2 package as well. For multi-sample patients, PCAs were produced with a matrix of SNP frequencies across all samples and the R's affycoretools package using the function "plotPCA" for the first two principal components. We also calculated the percentage of explained variation of these two principal components using the "prcomp" function.

**Prediction of WGS-DST profiles and comparison with culture DST.** Using the SNPs obtained from the analysis of genomic data, we performed an antibiotic resistance prediction for every sample. For this, reliable catalogs of resistance-associated variants were used, namely PhyResSE[56] and ReSeqTB[57] databases. We also considered as likely resistance variants any small INDEL present in genes commonly associated with antibiotic resistance. Once the predictions were obtained, we systematically compared them to the available phenotypic DST results to calculate matches and mismatches in the surgery patients dataset. We then computed sensitivity and specificity based on these coincidences and discrepancies.

**Identification of polyclonal infections.** Identification of polyclonal infections depends on the distribution of the different genotypes among patients samples (Fig. 1). When two genotypes are in two samples of the same patient, phylogenetic and genetic distances can be used to differentiate them. This happens mostly in superinfection cases where one strain prevails over the other. In contrast, when the different genotypes are in one sample, as it happens when both genotypes coexist, then deconvolution methods have to be applied to separate them before calculating their genetic distance. Consequently, we apply different methods to identify multiple strains depending on the number of samples available from a patient.

**Identification of polyclonal infections in single samples**
*Phylogenetic identification.* We reasoned that a single sample that shows evidence of two different genotypes in the same isolate should have two characteristics in a phylogeny. First, the sample should show no terminal branch length. Terminal branch lengths represent private fixed SNPs only present in the sample. When there is polyclonal infection no private SNPs are seen as any SNP not shared by the two genotypes will be at intermediate frequencies. The second rule is that they should not be part of a transmission cluster. It is known that transmission clusters are enriched in strains that are zero SNPs apart from other strains in the same cluster. Thus we required the candidate polyclonal infection to be at least 20 SNP apart from another sample in the dataset. We developed a Python script able to analyze a phylogeny, extract branch lengths using Python's ete3 toolkit, calculate phylogenetic distances between all samples, and identify terminal branches with 0 SNPs and more than 20 SNPs apart from any other isolate (Supplementary Fig. 4). For this, we generated a phylogeny using a neighbor-joining approach and Hamming distance, which represents branch length proportional to the absolute number of observed differences. Given the low genetic diversity of MTBC and that most positions are biallelic, Hamming distances reconstruct reasonably well the overall phylogeny with respect to maximum likelihood and it is easier to parse. To show the accuracy and the limit of detection of our approach we also generated in-silico mixes of strains with increasing genetic distance between the pairs selected (5, 10, 15, 20, 25 SNP) and added them to our phylogeny to test the script.

*Lineage markers identification.* To test the coexistence of two strains in one sample, we checked the appearance of any phylogenetic markers using a database from the literature[58,59] by means of a custom Python script. For samples that showed evidence of more than one marker from different MTBC lineages at a significant frequency (>5%), a polyclonal infection was called by this method and the estimated proportion of the involved genotypes was defined by the approximate frequencies of their markers.

*Deconvolution of individual genotypes.* For those cases in which we observed a polyclonal infection in just one sample by either the phylogenetic or lineage markers method, we established the proportions of the two strains in the sample and deconvoluted both the individual genotypes if their difference was big enough (e.g., 80/20 proportions, obtained by phylogenetic markers). By clustering the frequencies of all variants matching the phylogenetic markers' frequencies and assigning them to their corresponding genotype, we could isolate both and

calculate their genetic distance. For this, we developed an in-house Python script that uses a phylogenetic marker database from the literature[58,59] and takes the SNP file from the sputum sample to perform the detection and separation of both strains, computing their genetic distance solely based on the number of differences.

**Identification of polyclonal infections in multiple samples**
*Phylogeny manual inspection.* To identify polyclonal infections happening in two different samples from the same patient, we manually analyzed the phylogeny looking for isolates of the same patient that seat in different parts of the phylogeny. In addition, we recorded if samples were placed close enough to other patients samples in the phylogeny to suggest a recent reinfection event involving those two patients.

*Analysis of frequency spectra differences.* To make the detection process systematic for those cases where we had at least two samples (e.g., sputum1-sputum2 or sputum-caseum), we developed Python and R scripts to perform pairwise comparisons using SNP frequency differences obtained from the sample's genomic data, generating differences profiles that were plotted using ggplot2 package. We generated simple XY plots in which every point is a single variant determined by the frequencies in each of the samples involved in the comparison, and calculated $R^2$ as a measure of correlation (Supplementary Fig. 5). Polyclonal infections were almost always associated with low values of this index, except in cases where they were very subtle. We then proceeded to expand on the analysis generating a density graph that plots the distribution of SNP differences between the profiles. That is, in which range of frequencies is the comparison more enriched. This procedure allowed us to better define clonal infections when the enriched range was at the very low frequencies, and polyclonal infections when it was in the intermediate variable frequencies or at high fixed frequencies (Supplementary Fig. 6).

**Ethical approval.** Ethical approval for the study (#892/01-17) was obtained from the Institutional Review Board (IRB) of the National Center for Tuberculosis and Lung Diseases (NCTLD). The study design and conduct complied with all relevant regulations regarding the use of human study participants and was conducted in accordance with the criteria set by the Declaration of Helsinki. All enrolled patients or their legal guardians provided written informed consent prior to the inclusion in the study. All patient data were de-identified before final data analysis. All study staff during the study period had up-to-date GCP/GCLP certificates.

**Reporting summary.** Further information on research design is available in the Nature Research Reporting Summary linked to this article.

## Data availability
Sequencing files for all patients in the study are available at Bioproject accession codes PRJNA480888 and PRJNA318002. Individual accession codes for every sample are provided in Supplementary Data 1. All relevant data related to this work is available from the authors. Additional details about patients are also publicly available at TB Portals[60] (https://data.tbportals.niaid.nih.gov). Resistance prediction databases are available at PhyResSE (https://bioinf.fz-borstel.de/mchips/phyresse) and ReSeqTB (https://platform.reseqtb.org). Source data are provided with this paper.

## Code availability
Custom code used in this study, along with example inputs and outputs are available at https://gitlab.com/tbgenomicsunit/georgia-polyclonal (https://doi.org/10.5281/zenodo.4604579)[61]. The code is available under Creative Commons Attribution 4.0 International.

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

## Acknowledgements

The authors were supported by projects SAF2016-77346-R and PID2019-104477RB-I00 awarded to IC and the grant BES-2017-079656 awarded to MM by the Spanish Ministry of Economy and Competitiveness and Ministry of Science, the ERC project 638553-TB-ACCELERATE awarded to IC, Spanish Government-FEDER Funds through CV contract CPII18/00031 and grant PI16/01511, and Generalitat Valencia Grant to I.C. (code PROMETEO/2020/012). The grant providers played no part in study design, data collection, and analysis, or the preparation of the manuscript.

## Author contributions

I.C., M.M., S.V., N.S., Z.A., A.R., A.G., C.V., S.G., and R.K. contributed to the study design. N.S., I.K., Z.A., A.R., A.G., and S.V. collected or processed the data. SV performed the surgical resections. N.S., I.K., N.B., M.T., and L.V. performed microbiological or molecular biology work. I.C., M.M., N.S., S.G., R.K., and S.V. analyzed the data, which was interpreted by all authors. I.C., M.M., N.S., S.G., and R.K. drafted the manuscript with input from all authors. All authors have read and approved the final manuscript.

## Competing interests

The authors declare no competing interests.
