## [Peer Review File · Nature Communications]

REVIEWER COMMENTS

Reviewer #1 (Remarks to the Author):

This is an interesting and well-written manuscript about the diversity of *M. tuberculosis* in sputum samples and within different parts of cavitory lesions from patients subjected to surgery in the country of Georgia. The results are novel, providing insight on the pathogenesis of pulmonary tuberculosis.

Among the 18 patients with surgical samples, seven (39%) were found to have infections by two phylogenetically unrelated genotypes and some of them had strains that belong to different transmission clusters, suggesting super infection. None of the patients with surgical samples have polyclonal infection based on the sputum, although the time between sputum collection and the surgery was, in most of the cases, more than 2 months apart. The frequency of polyclonal infections in patients with only one sputum available and in patients with two sputum samples was 5% and 18%, respectively. Missing is information about the timing of the collection of these sputum samples relative to the use of anti-tuberculosis drugs, however most likely these sputum samples were obtained before the treatment.

The authors also found that the drug susceptibility profile in the surgical sample was different from the sputum samples in some of the patients.

The diversity within and around the surgical specimens also varied among patients with no polyclonal infection. Interestingly, the investigators found a high proportion of ROS-associated mutations in surgical samples compared with sputum samples, probably, as suggested by the authors, due to the antibiotic treatment and the host immune response

The authors concluded that the study of surgical lung samples provide a more complete picture of the within-patient diversity of *M. tuberculosis* compared to sputum samples, which could compromise the DST results. Because in several cases strains in transmission cluster were involved, the author suggested that either there are uncontrolled hotspots of transmission shared by MDR-TB patients, there is increased host susceptibility to reinfection by different genotypes, or both.

Finally, the authors commented that these results should be considered in the vaccine development as well as in the understanding in the studies regarding the immune and drug resistance selective pressure.

There are 2 issues that lessen the impact of this manuscript.

1. Because the sputum and surgical specimens were collected at different time points relative to treatment, the magnitude of the difference of the frequency of polyclonal infection should be considered cautiously.
2. Missing is information about the patients included in the different groups (one sputum tested, 2 sputum tested, and patients with sputum and surgical sample). For example, it will be important to know and control for use of anti-TB drugs before the sample collection when analyzing the association of polyclonal infection and type of sample (sputum vs lung). It is likely that the use of anti-TB drugs is a selective force for diversity within granulomas, which may explain some of the high frequency of polyclonal infection in this group. This analysis will strengthen the association of polyclonal infection and cavities.

Other observations:

Missing is the information about the procedures used to decrease the likelihood of cross-contamination between samples that may cause false low frequency variance.

Figure 1. The scenario in 1B is not a polyclonal infection. Because genotype 1 MTB was cleared in timepoint 1, the TB in timepoint 2 is only caused by genotype 2.

P5, first paragraph. Is this drug resistant profile from the initial sputum samples?

Figure 3C. Please revise the legends of the figure. If the legend is correct, then describe the dynamics of -shared, resistance and subpopulations-.

Page 12, third paragraph. The study is not designed to suggest that in high burden setting the first episode of TB might not protect against a second episode.

Page 15. Suggest including numerators and denominators for the percentage related to the main indication for lung surgery.

P17. Supplementary Table 2 (the list of discarded genomic features) is missing

P28. Supplementary table one. Please spell-out Prof

P26/ P30. The text mentions the principal component analysis for patient G018 (Supplementary Figure 2C). However, the supplementary figure 2c does not include this information.

Page 27. It seems premature to suggest that the proposed mutations explain PAS resistance with the studies described in this manuscript.

P32. Supplementary figure 5. Please describe the meaning of the 2 colors used in the graphs.

P33. Supplementary figure 6. Please describe what is the meaning of the different colors. Also, please describe what a negative density means. I suggest including an example of how to interpret one of the profiles.

Reviewer #2 (Remarks to the Author):

This is a relevant and rigorously conducted cohort study including 275 patients with tuberculosis, many of whom had multi-drug resistant disease. Key findings include the frequent discrepancy in polyclonal infections identified between sputum and lung tissue samples from the same individual, and the variable proportion of polyclonal infections within the lungs of different patients. This work extends current knowledge relating to the genotypic diversity of isolates within individuals with drug-resistant disease. The conclusions are supported by the evidence provided.

1. Could the authors please clarify processes in place to ensure was no laboratory cross-contamination during the study?
2. Do the authors have information about the treatment adherence of patients prior to surgery? Poorer adherence may explain an increased frequency of acquired resistance.
3. Do the authors have information about hospitalisation of these patients - as prolonged hospital exposure may create the opportunity for exogenous superinfection.
4. Is the HIV status of the patients known? This information would be useful, given the observation that other studies have shown a lower frequency of polyclonal infection in people with HIV.
5. Was there a correlation between the antibiotics used prior to sample collection and resistance of the isolates, or the proportion of isolates resistant to those antibiotics?
6. This paper provides useful insights into the potential variability of polyclonal infections within patients with drug-resistant disease. However, its findings are not readily generalisable given the specific eligibility requirements and consequent selection bias. The authors recognise this to some extent in the discussion. However, this could be explained further - this population is characterised

by (a) substantial exposure to multiple antibiotics, (b) a high frequency of prior TB disease, (c) the frequent presence of cavitory disease (associated with poor antibiotic penetration), and (d) likely exposure to exogenous superinfection during a prolonged hospitalisation in a drug-resistant TB ward. For this reason, the patient population is strongly selected for heteroresistance. The small number of patients with surgical samples is also limits generalisability.

7. An important additional consideration for future clinical trials and vaccine studies could be mentioned: The authors have shown that sputum is a poor correlate for lung polyclonal infection. Yet, this is the basis for distinguishing between relapse and reinfection in many clinical trials. This study gives rise to caution against the use of sputum samples as a method for ascertaining reinfection - with a low sensitivity and specificity of a single sputum sample for polyclonal infection. This additional comment could be incorporated in the discussion.

Minor issues

Figure 5: When was the timing of the surgery for patient G-031 - this dose not appear on the figure.

Reviewer #3 (Remarks to the Author):

In the manuscript "Genomic diversity of *M. tuberculosis* from human lung resections reveals a high frequency of polyclonal infections and impact of granuloma microenvironment" the authors present a higher frequency of polyclonal infection in surgically obtained material than could be identified from these patients sputa. The carefully sampled surgical specimens and their derived cultures allow for comparison of genetic diversity across the granuloma, also within one bacterial clone. This has to date only been performed on autopsy samples. The prospective cohort approach allows for estimations of the frequency of such events, enhancing generalizability. The bioinformatics approaches to deconvoluting different genotypes within one set of sequence reads is challenging yet seems robust, with in silico sequences generated and analysed as quality controls. The implications are that a single sputum derived drug resistance profile may underestimate the level of resistance in a patient. Moreover, this study suggests that- similar to protection from prior infection against re-infection seen in non-human primates- polyclonal infections with highly similar strains may be less common, with implications for vaccine immunity.

Major comments:

1. The 1st figure shows the possible variations in outcome of multiple infections.
 - a. Can the authors reflect in the discussion more on the timing of infection and implications for the understanding of the natural history of TB, also referring to experimental evidence from non-human primates?
 - i. What are the odds (scenario 1A) that two different genotypes are transmitted together? Maybe more likely shortly after another in a high risk (even nosocomial) setting? Does evidence exist, within this study or others, for patients linked in a transmission chain to have the same combination of two genotypes?
 - ii. If, in scenario 1C, infection G1 was controlled (patient asymptomatic), is it likely that the immune response triggered by G2 'wakes up' G1?
 - b. In the discussion, second paragraph, please refer back to Fig 1- superinfection may also present as mixed genotypes.
2. Can Fig 1 or S4 be expanded with the methodological cut-offs/ definitions for increased clarity? The bioinformatics approach to identifying polyclonal infection is well explained, yet also within one clone (?) vast number (>600) of (unfixed) SNPs seem to be identified, i.e. in excess of the 20 SNP cut-off to determine polyclonal infection (Fig4A).
3. Similarly, it is not always clear in figures what 'SNP frequency' is referred to, i.e. which denominator.

4. Administrative/ processing errors will yield different genotypes from multiple samples from the same patient. On the other hand, the concordance between mixed genotypes in different samples from the same patient could not be explained by error. While the renowned Tbilisi laboratory works according to GCLP with quality assurance in place, errors need to be excluded as thoroughly as possible.

- a. An additional useful quality control step would be to examine clustered isolates from the entire cohort by date/ time of sputum processing for instances of suspect laboratory cross contamination.
- b. P16: "Resected tissue samples were homogenized using Minilys homogenizer"- were negative controls included to exclude carry-over of tissue/DNA between samples?

5. ROS effect on transitions seen in caseum: as shown in Fig5, usual practice is that patients receive treatment before surgery. Could the increased transitions in caseum have resulted from antibiotic stress on MTB, not yet reflected in surgical sputum (not yet expectorated the centre of the cavity)? Can time between treatment start and caseum analysis be included as variable in the analysis?

Minor:

1. p2 Shorten into one sentence "However, individual sputum samples may underestimate the true bacterial diversity within the lung. Importantly, sputum samples are likely limited to reveal the coexistence of multiple *M. tuberculosis* strains in the same patient.
2. Table 1: add legend of C I E H N S
3. Fig 3B hard to see difference green and blue
4. p17 methods: DNA extraction: "plates were thoroughly scraped to maximise diversity recovery for bulk sequencing" versus "the CTAB/chloroform method was used for DNA extraction from 1mL of MGIT culture"- which approach was used? Was the difference between LJ and MGIT derived genomes (e.g. proportion of genotypes) from the same sample assessed as part of culture bias?
5. In the methods it is stated that a 3% cut-off was used for minority variant detection- best also explicitly state in the main body of the paper. The > 600 vSNPs found in G025 were then present in frequencies of 3-x%? See earlier comment on clarity in polyclonal vs clonal definitions.
 - a. FigS2B shows vSNPs <3%?
6. P13 discussion: "Even though we had access to surgery samples, analyses were done on cultured-samples and not directly on the surgery or sputum sample"- would it be possible to validate some of the findings by (target)sequencing directly on samples with sufficient AFB, to assess if even greater diversity was lost on culture?
7. Fig S5: blue vs red? Each dot is a specific vSNP?

Bouke de Jong

REVIEWER COMMENTS

Reviewer #1 (Remarks to the Author):

This is an interesting and well-written manuscript about the diversity of *M. tuberculosis* in sputum samples and within different parts of cavitory lesions from patients subjected to surgery in the country of Georgia. The results are novel, providing insight on the pathogenesis of pulmonary tuberculosis.

Among the 18 patients with surgical samples, seven (39%) were found to have infections by two phylogenetically unrelated genotypes and some of them had strains that belong to different transmission clusters, suggesting super infection. None of the patients with surgical samples have polyclonal infection based on the sputum, although the time between sputum collection and the surgery was, in most of the cases, more than 2 months apart. The frequency of polyclonal infections in patients with only one sputum available and in patients with two sputum samples was 5% and 18%, respectively. Missing is information about the timing of the collection of these sputum samples relative to the use of anti-tuberculosis drugs, however most likely these sputum samples were obtained before the treatment.

The authors also found that the drug susceptibility profile in the surgical sample was different from the sputum samples in some of the patients.

The diversity within and around the surgical specimens also varied among patients with no polyclonal infection. Interestingly, the investigators found a high proportion of ROS-associated mutations in surgical samples compared with sputum samples, probably, as suggested by the authors, due to the antibiotic treatment and the host immune response

The authors concluded that the study of surgical lung samples provide a more complete picture of the within-patient diversity of *M. tuberculosis* compared to sputum samples, which could compromise the DST results. Because in several cases strains in transmission cluster were involved, the author suggested that either there are uncontrolled hotspots of transmission shared by MDR-TB patients, there is increased host susceptibility to reinfection by different genotypes, or both.

Finally, the authors commented that these results should be considered in the vaccine development as well as in the understanding in the studies regarding the immune and drug resistance selective pressure.

There are 2 issues that lessen the impact of this manuscript.

1. Because the sputum and surgical specimens were collected at different time points relative to treatment, the magnitude of the difference of the frequency of polyclonal infection should be considered cautiously.

We are aware of this limitation in our study and it is described in the Discussion section. We tried to assess the effect of sampling time by comparing our sputum-surgery dataset with another sputum1-sputum2 dataset (Supp. Table 1) in which the time difference between samples is similar overall. By doing this we found that sputum pairs clearly improve the likelihood of detecting polyclonal infections (up to 18%), however it is still lower than the 39%

from the surgery dataset. This comparison suggests that surgical lung samples provide a more complete picture of the within-patient diversity of *M. tuberculosis*.

2. Missing is information about the patients included in the different groups (one sputum tested, 2 sputum tested, and patients with sputum and surgical sample). For example, it will be important to know and control for use of anti-TB drugs before the sample collection when analyzing the association of polyclonal infection and type of sample (sputum vs lung). It is likely that the use of anti-TB drugs is a selective force for diversity within granulomas, which may explain some of the high frequency of polyclonal infection in this group. This analysis will strengthen the association of polyclonal infection and cavities.

All information from patients in the different groups is available at Supp. Table 2, together with their TB Portals identifiers to consult additional data online. We agree with the reviewer that antibiotics can be a selective force for diversity within granulomas. Regardless of the group, baseline sputum samples at the time of collection did not have this drug pressure in most cases (excluding relapses) while surgical samples do. We recognise this limitation but unfortunately we cannot compare with healthy individual's surgical samples due to the nature of these samples.

Other observations:

Missing is the information about the procedures used to decrease the likelihood of cross-contamination between samples that may cause false low frequency variance.

We understand cross-contamination is a shared concern from all reviewers. These are the procedures or reasons why we think it is very unlikely:

1. Samples collection dates and processing dates from all patients are not close in time thus they haven't shared the same space. All this information is available at TB Portals.
2. Minilys homogenization is carried out in closed special, disposable tubes so that samples cannot be mixed.
3. Patients' genotypes match their DST phenotypes in nearly all cases, arguing against a general contamination problem
4. Not all polyclonal infections are in a transmission cluster and thus they don't match any other strain processed in the laboratory

Figure 1. The scenario in 1B is not a polyclonal infection. Because genotype 1 MTB was cleared in timepoint 1, the TB in timepoint 2 is only caused by genotype 2.

While we agree with the reviewer that this scenario is not technically a polyclonal infection but two TB episodes, what we are trying to convey with this figure are the possible scenarios that can result in two genotypes across two or more samples. Given the time differences between samples collection, we cannot know for sure what happened in the middle, but the depicted scenarios cover the most likely events that explain what we see retrospectively. In reality many of these scenarios will never be teased out for each patient. For example we will never be sure if two strains were already infecting at baseline while only one is detected in sputum as we don't have access to lung samples before treatment. This is true for this

study and for any other study in TB. We have fleshed out this point in the Discussion section, second paragraph.

P5, first paragraph. Is this drug resistant profile from the initial sputum samples?

The shown percentages do not represent patients profile but samples profile. We chose to do this precisely due to the differences in within-patient profile.

Figure 3C. Please revise the legends of the figure. If the legend is correct, then describe the dynamics of -shared, resistance and subpopulations-.

We have updated the figure changing 'subpopulations' to 'unassigned' for better clarity and added explanatory text to the figure legend for each population dynamic.

Page 12, third paragraph. The study is not designed to suggest that in high burden setting the first episode of TB might not protect against a second episode.

We agree that it is not designed for that but it is an observation derived from the main result. As suggested elsewhere we acknowledge that this is a special population which may be more susceptible to reinfection. We concur with the reviewer in that we would need a case-control study to test that specifically.

We have now recognized this in the paragraph mentioned by the reviewer: 'However, our results suggest that in a high-burden setting this might not be the case and future studies designed to corroborate this observation will be needed.'

Page 15. Suggest including numerators and denominators for the percentage related to the main indication for lung surgery.

We have introduced this change in the new version.

P17. Supplementary Table 2 (the list of discarded genomic features) is missing

The list of discarded genomic features is Supp. Table 3, which was included in the original submission. There was a numbering mistake in the Methods section that has been corrected.

P28. Supplementary table one. Please spell-out Prof

We have changed it to 'Profile' in the new version.

P26/ P30. The text mentions the principal component analysis for patient G018 (Supplementary Figure 2C). However, the supplementary figure 2c does not include this information.

We thank the reviewer for spotting this missing information. The figure has been updated in the new version of the manuscript and now includes all patients PCA plots with an added legend.

Page 27. It seems premature to suggest that the proposed mutations explain PAS resistance with the studies described in this manuscript.

We have changed the wording of the statement to be more hypothetical. Further studies would be needed to assess the role of these mutations.

P32. Supplementary figure 5. Please describe the meaning of the 2 colors used in the graphs.

We have added to the figure legend that 'blue graphs correspond to clonal infections and red graphs to polyclonal infections'.

P33. Supplementary figure 6. Please describe what is the meaning of the different colors. Also, please describe what a negative density means. I suggest including an example of how to interpret one of the profiles.

Supplementary Figure 6 has been updated. We have added a second panel with 3 examples explaining how to interpret the density graphs. In addition, the legend now reflects what the colors represent.

Reviewer #2 (Remarks to the Author):

This is a relevant and rigorously conducted cohort study including 275 patients with tuberculosis, many of whom had multi-drug resistant disease. Key findings include the frequent discrepancy in polyclonal infections identified between sputum and lung tissue samples from the same individual, and the variable proportion of polyclonal infections within the lungs of different patients. This work extends current knowledge relating to the genotypic diversity of isolates within individuals with drug-resistant disease. The conclusions are supported by the evidence provided.

1. Could the authors please clarify processes in place to ensure was no laboratory cross-contamination during the study?

Please see Reviewer 1 answer about the same topic.

2. Do the authors have information about the treatment adherence of patients prior to surgery? Poorer adherence may explain an increased frequency of acquired resistance.

This information is available for all patients at TB Portals (identifiers at Supp. Table 2). Treatment adherence was overall very high so we discard this as a reason for increased resistance acquisition in this group. We find more likely that the problem in the setting is transmission of already resistant strains as highlighted by the fact that many polyclonal infections match other strains in the dataset.

3. Do the authors have information about hospitalisation of these patients - as prolonged hospital exposure may create the opportunity for exogenous superinfection.

We have reviewed the medical histories of the surgical group and while some of them had a more prolonged hospital exposure than others, they don't always match polyclonal infection cases, so we cannot correlate these two facts right now. However, we agree that it is likely that prolonged hospitalization and thus exposure can greatly increase the risk of superinfection. We have added this to the discussion as we think it is good for the reader to consider all possibilities

4. Is the HIV status of the patients known? This information would be useful, given the observation that other studies have shown a lower frequency of polyclonal infection in people with HIV.

This information is available online at TB Portals for every patient in the study along with other epidemiological data (identifiers found in Supplementary Table 2), but unfortunately the number of HIV+ individuals is too low to perform any statistical analysis. Only 3 out of the 275 patients were HIV+, and none of those 3 showed polyclonal infection.

5. Was there a correlation between the antibiotics used prior to sample collection and resistance of the isolates, or the proportion of isolates resistant to those antibiotics?

In most cases, baseline sputum isolates were already resistant to several antibiotics and did not acquire new resistances over the course of treatment and up to surgical resection. However, we found some cases that developed new resistances during treatment, like G019 or G085 acquiring ofloxacin resistance. These instances are shown on Supplementary Figure 7.

6. This paper provides useful insights into the potential variability of polyclonal infections within patients with drug-resistant disease. However, its findings are not readily generalisable given the specific eligibility requirements and consequent selection bias. The authors recognise this to some extent in the discussion. However, this could be explained further - this population is characterised by (a) substantial exposure to multiple antibiotics, (b) a high frequency of prior TB disease, (c) the frequent presence of cavitory disease (associated with poor antibiotic penetration), and (d) likely exposure to exogenous superinfection during a prolonged hospitalisation in a drug-resistant TB ward. For this reason, the patient population is strongly selected for heteroresistance. The small number of patients with surgical samples is also limits generalisability.

We thank the reviewer for this valuable insight on the study population limitations. We have expanded the limitations paragraph in the Discussion section to reflect his/her concerns.

7. An important additional consideration for future clinical trials and vaccine studies could be mentioned: The authors have shown that sputum is a poor correlate for lung polyclonal infection. Yet, this is the basis for distinguishing between relapse and reinfection in many clinical trials. This study gives rise to caution against the use of sputum samples as a method for ascertaining reinfection - with a low sensitivity and specificity of a single sputum sample for polyclonal infection. This additional comment could be incorporated in the discussion.

We have added a new sentence to the Discussion section regarding this as suggested.

Minor issues

Figure 5: When was the timing of the surgery for patient G-031 - this dose not appear on the figure.

We thank the reviewer for spotting this missing data point. It has been added in the new version.

Reviewer #3 (Remarks to the Author):

In the manuscript 'Genomic diversity of *M. tuberculosis* from human lung resections reveals a high frequency of polyclonal infections and impact of granuloma microenvironment' the authors present a higher frequency of polyclonal infection in surgically obtained material than could be identified from these patients sputa. The carefully sampled surgical specimens and their derived cultures allow for comparison of genetic diversity across the granuloma, also within one bacterial clone. This has to date only been performed on autopsy samples. The prospective cohort approach allows for estimations of the frequency of such events, enhancing generalizability. The bioinformatics approaches to deconvoluting different genotypes within one set of sequence reads is challenging yet seems robust, with in silico sequences generated and analysed as quality controls. The implications are that a single sputum derived drug resistance profile may underestimate the level of resistance in a patient. Moreover, this study suggests that- similar to protection from prior infection against re-infection seen in non-human primates- polyclonal infections with highly similar strains may be less common, with implications for vaccine immunity.

Major comments:

1. The 1st figure shows the possible variations in outcome of multiple infections.
 - a. Can the authors reflect in the discussion more on the timing of infection and implications for the understanding of the natural history of TB, also referring to experimental evidence from non-human primates?

We have now added a new paragraph to the Discussion section covering this topic more in-depth, along with question 1.a.ii. There is data in the literature suggesting that it may be the case that a secondary infection drives progression of a primary infection. Unfortunately it is a scenario we cannot test with the current setup.

- i. What are the odds (scenario 1A) that two different genotypes are transmitted together? Maybe more likely shortly after another in a high risk (even nosocomial) setting? Does evidence exist, within this study or others, for patients linked in a transmission chain to have the same combination of two genotypes?

The odds of scenario 1A are likely to be very low, but still possible. As the reviewer suggests, the most plausible explanation in several cases is that superinfection occurred. We cannot classify most of the patients but these theoretical scenarios cover most of the

situations. Unfortunately, we haven't found any case of two patients with the same combination of genotypes in a transmission cluster in this study.

ii. If, in scenario 1C, infection G1 was controlled (patient asymptomatic), is it likely that the immune response triggered by G2 'wakes up' G1?

We thank the reviewer for this insightful comment on a possibility that was not contemplated before in the discussion section and has now been added (see 1.a).

b. In the discussion, second paragraph, please refer back to Fig 1- superinfection may also present as mixed genotypes.

We have included the reference to Figure 1C in this paragraph in the new version.

2. Can Fig 1 or S4 be expanded with the methodological cut-offs/ definitions for increased clarity? The bioinformatics approach to identifying polyclonal infection is well explained, yet also within one clone (?) vast number (>600) of (unfixed) SNPs seem to be identified, i.e. in excess of the 20 SNP cut-off to determine polyclonal infection (Fig4A).

The 20 SNP cut-off to determine polyclonal infection is based on fixed variants, as these are the ones that define the phylogeny. We have now remarked this in Supp. Figure 4.

3. Similarly, it is not always clear in figures what \square SNP frequency \square is referred to, i.e. which denominator.

We have changed some figure labels to 'SNP frequency in culture' when it refers to the frequency of the variant in the population to make it clearer for a wider range of readers.

4. Administrative/ processing errors will yield different genotypes from multiple samples from the same patient. On the other hand, the concordance between mixed genotypes in different samples from the same patient could not be explained by error. While the renowned Tbilisi laboratory works according to GCLP with quality assurance in place, errors need to be excluded as thoroughly as possible.

a. An additional useful quality control step would be to examine clustered isolates from the entire cohort by date/ time of sputum processing for instances of suspect laboratory cross contamination.

Please see Reviewer 1 answer about the same topic.

b. P16: 'Resected tissue samples were homogenized using Minilys homogenizer'- were negative controls included to exclude carry-over of tissue/DNA between samples?

Please see Reviewer 1 answer about the same topic.

5. ROS effect on transitions seen in caseum: as shown in Fig5, usual practice is that patients receive treatment before surgery. Could the increased transitions in caseum have resulted from antibiotic stress on MTB, not yet reflected in surgical sputum (not yet expectorated the

centre of the cavity)? Can time between treatment start and caseum analysis be included as variable in the analysis?

We think both are compatible. Antibiotic pressure can be a contributor for the difference seen in transitions in surgical samples versus baseline sputa. We analyzed the effect of time using the paired sputa dataset and found that, while treatment and/or time did have an effect on this increase, the magnitude for surgical samples was greater suggesting a major role of ROS and immune pressures. In fact at the end of the discussion we propose a potential link between ROS and antibiotic pressure as transitions are involved in both phenomena. In some patients, differences are driven by a mutational signature associated with ROS and by treatment and suggest a link between immune and drug resistance selective pressures.

Minor:

1. p2 Shorten into one sentence 'However, individual sputum samples may underestimate the true bacterial diversity within the lung. Importantly, sputum samples are likely limited to reveal the coexistence of multiple *M. tuberculosis* strains in the same patient.'

We have combined the sentences in the updated new version.

2. Table 1: add legend of C I E H N S

We have added the legend to this table as suggested.

3. Fig 3B hard to see difference green and blue

We have updated this figure's colors for clarity.

4. p17 methods: DNA extraction: 'plates were thoroughly scraped to maximise diversity recovery for bulk sequencing' versus 'the CTAB/chloroform method was used for DNA extraction from 1mL of MGIT culture'- which approach was used? Was the difference between LJ and MGIT derived genomes (e.g. proportion of genotypes) from the same sample assessed as part of culture bias?

Both surgical and sputum samples were cultured on MGIT and LJ in parallel. The extraction protocols for each method are indeed somewhat different and we only used MGIT in cases where there was no LJ sufficient growth, although we believe thorough LJ plate scraping to be comparable to MGIT direct extraction in terms of diversity recovery. We have updated the Methods section detailing this for better clarity.

5. In the methods it is stated that a 3% cut-off was used for minority variant detection- best also explicitly state in the main body of the paper. The > 600 vSNPs found in G025 were then present in frequencies of 3-x%? See earlier comment on clarity in polyclonal vs clonal definitions.

We have added the 3% cut-off statement at the end of the Introduction section. As for patient G025, the mentioned variants are at 3-5% (Supp. Figure 3).

a. FigS2B shows vSNPs <3%?

It illustrates a set of SNPs that are originally above 3% frequency in the caseum sample and how we can trace their frequencies across the rest of the lesion. As we validate those variants as not spurious, it is highly likely that their detection in samples below 3% is accurate. We have added this explanation to the figure text.

6. P13 discussion: 'Even though we had access to surgery samples, analyses were done on cultured-samples and not directly on the surgery or sputum sample'- would it be possible to validate some of the findings by (target)sequencing directly on samples with sufficient AFB, to assess if even greater diversity was lost on culture?

The argument for diversity loss in culture can be made for every type of sample, and not only in this study. Target sequencing of some genes, however, is not straight-forward, given that it would not generally allow to identify polyclonal infections. Although direct WGS of surgical samples is out of our scope right now, we will be working on this in the future.

7. Fig S5: blue vs red? Each dot is a specific vSNP?

Blue-dotted graphs correspond to clonal infections and red-dotted graphs to polyclonal infections. Each dot is a single SNP, regardless of its frequency.

REVIEWERS' COMMENTS

Reviewer #1 (Remarks to the Author):

The authors responded to all the comments. However, the investigators didn't include in the text any comment about cross contamination (mentioned by the 3 reviewers) which I think should be included in the manuscript.

Reviewer #2 (Remarks to the Author):

The authors have adequately addressed the concerns raised in the previous review. Acceptance for publication is recommended.

Reviewer #3 (Remarks to the Author):

The authors adequately addressed comments made by reviewers.
Bouke de Jong

REVIEWERS' COMMENTS

Reviewer #1 (Remarks to the Author):

The authors responded to all the comments. However, the investigators didn't include in the text any comment about cross contamination (mentioned by the 3 reviewers) which I think should be included in the manuscript.

We thank reviewer 1 for their input and improving the quality of this manuscript. Also, we have now included the following text in the Discussion section to address the issue of cross-contamination:

'Finally, a possible limitation of this study would be cross-contamination as an explanation to the high frequency of polyclonal infections, although we think it is highly unlikely for a number of reasons. First, sample collection dates and processing dates from all patients are not close in time thus they haven't shared the same space. Second, sample homogenization is carried out in closed special, disposable tubes so that samples cannot be mixed. Third, genotypes match their DST phenotypes in nearly all cases, arguing against a general contamination problem. Lastly, not all polyclonal infections are in a transmission cluster and thus they don't match any other strain processed in the laboratory.'

Reviewer #2 (Remarks to the Author):

The authors have adequately addressed the concerns raised in the previous review. Acceptance for publication is recommended.

We thank reviewer 2 for their input and improving the quality of this manuscript.

Reviewer #3 (Remarks to the Author):

The authors adequately addressed comments made by reviewers.
Bouke de Jong

We thank reviewer 3 for their input and improving the quality of this manuscript.